# Strength Conditioning Program to Prevent Adductor Muscle Strains in Football: Does it Really Help Professional Football Players?

**DOI:** 10.3390/ijerph17176408

**Published:** 2020-09-02

**Authors:** Javier F. Núñez, Ismael Fernandez, Alberto Torres, Sergio García, Pablo Manzanet, Pascual Casani, Luis Suarez-Arrones

**Affiliations:** 1Department of Sports and Informatics, Sport Faculty, University of Pablo de Olavide of Sevilla, 41013 Sevilla, Spain; ljsuamor@upo.es; 2Physical Trainer Coach of Valencia C.F, 46010 Valencia, Spain; ismaelfernandezrodriguez@hotmail.com (I.F.); sergiogarciagarcia23@gmail.com (S.G.); 3Readapter of Valencia C.F., 46010 Valencia, Spain; albertotc84@gmail.com; 4Physical Trainer Coach of Villarreal C.F., 12540 Vila-Real, Spain; pablo_manzanet@yahoo.es; 5Head of Medical Staff of Valencia C.F., 46010 Valencia, Spain; casani@comv.es

**Keywords:** prevention, soccer (football), rotatory inertial device, adductor injury, unilateral deficits, ratio adductor-abductor

## Abstract

Coaches at the professional level are often concerned about negative side effects from testing and intensive resistance training periods, and they are not willing to base their training prescriptions on data obtained from semiprofessional or amateur football players. Consequently, the purpose of this study was to analyze the reliability and effectiveness of two adductor injury active prevention programs using the adductor/abductor ratio and deficit between legs, on the basis of adduction–abduction power output during the exercises proposed, in professional football players. Forty-eight professional football players undertook complementary strength training for the adductor and abductor muscles in their dominant and non-dominant legs, once or twice a week throughout the playing season. The volume of the session was determined by the adductor/abductor ratio and the deficit between legs in the last session training measured. The number and severity of muscle injuries per 1000 h of exposure were recorded. Both prevention programs showed a very low rate of adductor injury (0.27 and 0.07 injuries/1000 h) with mild-to-moderate severity, maintaining a balance in percentage asymmetry between dominant and non-dominant legs for adductor (10.37%) and in the adductor/abductor ratio (0.92) in top professional football players throughout the season. The strength conditioning program proposed can help to prevent adductor muscle injuries in top professional football players.

## 1. Introduction

Groin injuries are especially common in team sports [1]. In professional football players, 35% of groin injuries are located in the adductor muscles, and 29% of adductor injuries are subsequently reinjured [2]. The adductor muscle injury incidence is estimated to be between 0.8 [3] and 0.4 injuries per 1000 h of participation [4]. Specifically, in First Division Spanish football, hip muscle injury incidence is 0.61 injuries per 1000 h of exposure, and there is a mean of 8.6 days for recovery [5]. This incidence of adductor injuries in football requires the design of prevention programs that are more effective than those developed thus far.

Designing an active prevention program requires identification of the risk of adductor injury, followed by the choice of an active program of specific exercises that improve the coordination of the muscles acting on the pelvis [6]. Adductor strength testing is an important tool for identifying players who are at risk of adductor injury [7]. There are different methods of hip muscle strength testing in football players: isometric strength dynamometry [8,9,10,11], eccentric strength dynamometry [9,12], and isokinetic dynamometry [13]. To date, adductor injury recovery factors have been suggested to include (a) an adductor/abductor ratio of more 90% [10,12,14], (b) a lower extremity strength deficit of more than 10% on the injured compared with the uninjured leg [1,12], and (c) a reduced hip adduction force with respect to the last measured force [10,14,15,16,17]. Therefore, an injury prevention program should include exercises for hip adduction–abduction strengthening and testing in order to decrease the stress on the adductor muscle–tendon unit to prevent overuse injury [9].

Interventions with elite level football players present evident problems. Training interventions using randomized controlled trials are impractical and/or ethically questionable in elite populations [18]. Coaches at the elite level are often concerned about the possible risks of strength testing and intensive training periods, including fatigue, reduced trainability, or possible future muscle injury [18,19]. In addition, coaches are not willing to base their training prescriptions on data obtained from semiprofessional or amateur football players [18]. To our knowledge, there are no data on elite football players that give reference values and that inform us about the necessary ratios, asymmetries, and how these possible risk factors can be modified during training to produce better-balanced players, minimizing the possible risk of injury. An alternative to testing the hip muscle strength in professional football players could be to measure the adductor–abductor muscle profiles during specific exercises in a prevention program designed to reduce injury. Consequently, the purpose of this study was to analyze the effectiveness of two adductor injury-prevention programs, using the adductor/abductor ratio and the deficit between legs, on the basis of adduction–abduction power output during exercises in two proposed programs designed for professional football players. It was hypothesized that both programs are reliable and effective in preventing adductor injury in professional football players during a full season.

## 2. Materials and Methods

### 2.1. Experimental Approach to the Problem

In this investigation, a cohort study was used to analyze the effectiveness of two exercise programs, on the basis of a functional power test, in preventing adductor muscle strains. Professional football players were investigated to analyze the influence of these training programs on the number of adductor injuries sustained during a full season. The study-dependent variables included mean concentric power, unilateral deficits between the dominant and non-dominant legs, the adductor/abductor ratio, the number of muscle injuries per 1000 h of exposure (match and training), and the severity of the injuries.

### 2.2. Participants

Forty-eight professional football players with more than 3 years of experience in professional football participated in this study. The subjects’ mean ± SD age, height and body mass were 25.1 ± 3.5 years, 178 ± 4.6 cm, and 76.4 ± 3.6 kg, respectively. These data were obtained from daily monitoring during strength training. Therefore, the usual ethics committee clearance was not required for this aspect [20]. The study was approved by the ethics and research committee of the Virgen Macarena and Virgen del Rocío university hospitals (0398-N-17) and was performed in accordance with the ethical standards laid down in the 1964 Declaration of Helsinki and its later amendments or comparable ethical standards.

The participants were recruited from two professional football teams that competed in the first division of the Spanish football league (La Liga) during the season in which each team was evaluated. Initially, excluding the goalkeepers, each team was made up of 24 players (TEAM A, TEAM B). Ten players in TEAM A and 11 in TEAM B were excluded because they suffered injuries unrelated to those evaluated in this study, which did not allow them to follow the work dynamics. No player excluded from this study suffered an adductor injury during the full season.

### 2.3. Hip Adduction/Abduction Isometric Strength

Before the beginning of the preseason, isometric maximum voluntary contraction (MVC) was measured by medical staff for all football players, over a period of 5 s in each case, to evaluate adduction/abduction strength in the dominant and nondominant legs. All test procedures were standardized and performed in a supine position as described by Thorborg [21], using a portable handheld dynamometer (Lafayette Instrument Company, Lafayette, LA, USA; accuracy: ± 0.88N; reliability: ICC: 0.76 (0.24–0.94); CV: 8.7% and 0.89 (0.57–0.97), CV: 7.2% for hip-abduction and adduction, respectively [21]) that was calibrated before testing. The test was conducted four times for each individual, with a 30-s rest period between each trial, and the highest value was selected for further analysis.

### 2.4. Hip Adduction/Abduction Power during Dynamic Assessement

Adduction (ADD) and abduction (ABD) power were measured using a rotatory inertial device (conic-pulley; Twister Pulley: Element Sport, Cádiz, Spain; speed/force ratio level 1 out of 3). These systems provide a source of linear resistance via a tether wrapped around a vertical cone-shaped shaft [22]. The kinetic energy from the acceleration phase of the exercise is transferred to the deceleration phase, where an equal impulse is necessary to halt the rotation of the moment of inertia [23]. This exercise device uses the inertia of a spinning flywheel (moment of inertia = 0.11 kg m^−2^). An encoder attached to the rotation axis of the conic-pulley was connected to hardware (SmartCoach, SmartCoach Europe AB, Stockholm, Sweden) with associated SmartCoach software (SmartCoach v.5.2.0.5) to measure the power (w) for each repetition [24]. The SmartCoach Pro system (SmartCoach, SmartCoach Europe AB, Stockholm, Sweden) was developed to monitor several football players using six conic-pulleys simultaneously. Power data from the conic-pulley was automatically stored in the Cloud (a central database) for further analysis.

After a standardized warm-up, the subjects completed three sets of eight repetitions using a conic-pulley for the ADD dominant leg (ADD_D_), ADD non-dominant leg (ADD_ND_), ABD dominant leg (ABD_D_), and ABD non-dominant leg (ABD_ND_). The means of the three best repetitions (mean_power (w)) during the concentric phase of the ADD and ABD exercises were selected for further analysis. The rest time between sets and between exercises was 3 min. The asymmetries between dominant and non-dominant legs were expressed as a percentage (%Asy) and calculated as (dominant − non-dominant)/non-dominant [12]. The adductor/abductor ratio (ADD:ABD ratio) was expressed as a percentage and calculated as ADD·ABD^−1 21^.

### 2.5. Training Intervention

The football players performed a complementary ADD/ABD strength training twice a week for 8 weeks in the pre-season and once a week during 37 in-season weeks. The first 2 weeks of the preseason were used as a familiarization period, and after this period the mean_power, %Asy, and ADD/ABD ratios were calculated during the strength training sessions. From the third week of the pre-season until the end of the season, power, %Asy, and ADD/ABD ratios were measured every 6 weeks (i.e., weeks 3, 9, 15, 21, 27, 33, and 39).

The subjects were placed in the supine position [10], perpendicular to the conic-pulley, and stabilized themselves with their hands on the floor. The conic-pulley rope was fixed at the lower end of the leg, so that the axis of the rope was aligned with the ADD–ABD of the hip for all ranges of movement (Figure 1 and Figure 2). During a monitoring training session, the subjects completed three sets of two initial repetitions to accelerate the conic-pulley and eight repetitions applying maximal effort for ADD_D_, ADD_ND_, ABD_D_, and ABD_ND_.

The training volume for ADD and ABD between monitoring sessions was determined by the last %Asy and ADD/ABD ratio obtained. The adductor injury prevention program for TEAM A was based on a reduced volume (i.e., number of sets) for the leg with the higher power output, and in TEAM B was based on an increased volume for the leg with the lower power output (see Table 1).

### 2.6. Muscle Injuries and Exposure

Injury information was classified by the medical staff and a recordable injury was defined as one that caused an absence from future football participation [25]. Medical staff recorded the exposure to training sessions and matches and registered the number of muscle injuries per 1000 h. A player was considered fully rehabilitated when the medical team allowed full participation in team training or matches [26]. Following Hägglund et al. [2], injuries were categorized under the following four degrees of severity, on the basis of layoff time from football: slight/minimal (0–3 days), mild (4–7 days), moderate (8–28 days), and severe (>28 days). A recurrent injury was defined as an injury of the same type and at the same site as an index injury, occurring after a player’s return to full participation [25].

### 2.7. Statistical Analysis

Descriptive statistics were calculated for each variable. Data distribution was examined for normality using the Shapiro–Wilk test. Pearson correlation (±95% CI) was used to compare isometric MVC with functional measures. Correlation coefficients were qualitatively ranked by magnitude as follows: trivial, *r* < 0.1; small, 0.1 < *r* < 0.3; moderate, 0.3 < *r* < 0.5; large, 0.5 < *r* < 0.7; very large, 0.7 < *r* < 0.9; almost perfect, 0.9 < *r* < 1.0; and perfect *r* = 1.0 [27]. Test–retest reliability was assessed using the standard error of measurement as a coefficient of variation (%CV) [28], and the intra-class correlation coefficient (ICC) with a 95% CI. A repeated-measures design, using a one-way within-subjects ANOVA test, followed by Bonferroni post hoc tests, was used to investigate differences in variables measured within groups. A *t*-test was used to investigate differences in variables measured between groups. The effect sizes (ES) of the standardized differences within groups and between groups were determined using Hedges’ g statistic, and Hopkins’ scale was used to determine the magnitude of the effect size, where 0–0.2 = trivial, 0.2–0.6 = small, 0.6–1.2 = moderate, 1.2–2.0 = large, and >2.0 = very large [27].

## 3. Results

Hip ADD/ABD exercises showed a good absolute and relative reliability for all measures: ICC: 0.92 (0.86–0.96); CV: 7.1%. There were no relationships between the isometric MVC and the power obtained during the dynamic assessment in the inertial device for ADD_D_, ADD_ND_, ABD_D_, and ABD_ND_.

Within-group analyses of consecutive monitoring training sessions showed a significant interaction for ADD_D_ power, ADD_ND_ power, ABD_D_ power, ABD_ND_ power (*F* = from 8.647 to 11.428; *p* < 0.001). Post hoc analyses showed that TEAM A exhibited a significant reduction in ADD_D_ power (ES = 3.43, *p* < 0.01), ADD_ND_ power (ES = 1.52, *p* < 0.01), ABD_D_ power (ES = 3.65, *p* < 0.01), and ABD_ND_ power (ES = 2.20, *p* < 0.01) between the third (week 14) and fourth (week 20) monitored training sessions (see Figure 3).

Both of the adductor strength training programs showed no significant differences in ADD/ABD ratios, ADD %Asy, and ABD %Asy during the seven monitored training sessions and between different programs (see Figure 4 and Figure 5).

The adductor injury rate during training and matches was 0.09 ± 0.02 and 0.18 ± 0.03, respectively, for TEAM A, while for TEAM B it was 0.0 during training and 0.07 for matches. Soccer absenteeism was 11.33 ± 2.4 days (moderate severity on average) for TEAM A and 6.0 ± 0.0 days (mild severity on average) for TEAM B. All injuries occurred in the dominant leg between monitored sessions 4 (week 21) and 5 (week 27) for TEAM A, and between monitored sessions 2 (week 9) and 3 (Week 15) for TEAM B. None of the players suffered a reinjury.

## 4. Discussion

The purpose of this study was to analyze the effectiveness of two adductor strength training programs on the basis of ADD/ABD ratios and %Asy between legs in professional football players. The main findings of our study were (a) both strength training programs showed a low adductor injury rate (0.27 injuries/1000 h for TEAM A; 0.07 injuries/1000 h for TEAM B) in elite professional football players, (b) both strength training programs maintained a good balance in %Asy (10.37%) and for ADD/ABD ratios (0.92) for dominant and non-dominant legs, and (c) we did not find that decreased ADD power was a risk factor for subsequent adductor injuries in either training program.

In the current study, assessment of both injury prevention programs showed adductor injury rates of 0.27 and 0.07 injuries per 1000 h of exposure, with soccer absenteeism of 11.3 and 6 days in TEAM A and TEAM B, respectively. The observed injury rates were lower than those obtained by Noya et al. [5] with football players from the Spanish Football League (0.61 injuries/1000 h), although the soccer absenteeism (8.6 days) had a lower value than TEAM A, while being slightly higher than the absenteeism score for TEAM B. One possible explanation for these substantial differences in injury rates between studies is that Noya et al. [5] refer to hip/groin muscle injuries, including other muscles in addition to ADD (such as the psoas), which may have resulted in a greater injury rate in comparison with our results. However, the values presented by Hölmich et al. (0.4 injuries/1000 h) [4] for groin injuries in sub-elite football players were also lower than Noya et al., although higher than the results in the present study. In line with previous studies [2,5,29,30], our injury rate during matches was higher than during training, and the ADD injury rate in both teams (A and B) was lower than that found by Ekstrand et al. [29] for professional football players (0.32 injuries/1000 h of training; 2 injuries/1000 h during matches). These significant differences between competition and training are due to the greater volume and intensity of movement during competition, which induces a higher risk of injury [5]. Our results showed that all recorded injuries occurred in the dominant leg and none of the players experienced a reinjury. These data are not in accordance with the results obtained by Ekstrand et al. [29], which indicate that there is the same possibility of an injury in the dominant (54%) as in the non-dominant leg (46%). It is likely that the low injury rate showed in the present study could have contributed to the absence of injuries in the non-dominant leg.

During the season, both programs appeared to be effective in maintaining an optimal balance in %Asy, as assessed with the ADD and ABD exercises over the season. These results are in line with previous studies in which the isometric hip test was measured in the same position as our test [10] and in a side-lying position for hip abduction–adduction measurements [13]. The %Asy obtained in TEAM B was below 10% during the whole season, except in the second monitored training session (10.6%). Coincidentally, the only adductor strain injury for TEAM B occurred between the second (week 9) and third (week 15) monitored training sessions. Despite there being no significant differences between the two proposed protocols, the %Asy obtained in TEAM A was above 10% during the whole season, except in the fifth monitored training session (see Figure 4). Adductor strain injuries for TEAM A occurred between the fourth (week 21) and fifth (week 27) monitored training sessions. In both programs, injuries to the adductor muscles occurred around the time that the %Asy was above 10% in the previous measure, and thus this factor appears to be a good identifier of the risk of adductor injury. The TEAM B protocol seemed to offer fewer fluctuations in %Asy throughout the season than were observed in TEAM A.

An imbalance between the strength of the adductor and abductor muscles is a risk factor for groin injuries [12,13,14]. The current study showed that both individualized intervention programs maintained similar ADD/ABD ratios for the dominant and non-dominant legs (0.92) throughout the season. The hip abduction power was greater than the hip adduction power for both dominant and non-dominant legs, except in the second and fifth measures for the dominant leg in TEAM A. Our results from the dynamic assessment in the inertial device contrast with those obtained by Jensen et al. (2014), who showed that isometric hip adduction strength was greater than isometric hip abduction strength for both dominant and non-dominant legs in semi-professional football players (i.e., adductor/abductor ratios of 1.27) [9]. In line with Jensen et al. (2014), Poulmedis [30] (1985) reported higher isokinetic peak torque values in the hip adductors than in the hip abductors in football players (i.e., adductor/abductor ratio of 1.35). However, along the same line as our results, Belhaj et al. [13] showed that the isokinetic hip abductor strength was greater in all measurements than the isokinetic hip adductor strength in football players with and without chronic adductor-related groin pain. The lower adductor/abductor ratios were in the affected dominant (0.33) and non-dominant (0.62) legs, in comparison with the unaffected dominant (0.84) and non-dominant (0.81) legs. Our results at the beginning of the preseason were very similar to those obtained by Belhaj et al. [13] for the dominant and non-dominant legs; however, during the season, the individualized intervention program used with TEAM A alternated ADD/ABD ratios above and below 0.90, while TEAM B always maintained this ratio below 0.90. A possible explanation for these results could be the effect of individualized workload based on the %Asy and ADD/ABD ratios obtained. The adductor strength training program in TEAM A was based on a reduced volume (i.e., number of sets) in the leg with the higher power output, allowing players not to train ADD, ABD, or both muscles of one leg over 5 weeks. By contrast, this never happened in the preventive program used with TEAM B. On the basis of this and in order to maintain a stable ADD/ABD ratio, we should always train both legs, although with different volumes of training for each leg. In the case of TEAM A, the adductor injuries occurred between the fourth measured session (week 21) and the fifth (week 27), indicating that an ADD/ABD ratio of more than 0.9 could be a variable to consider to prevent adductor injury in football players [10,12,14]. In the case of TEAM B, the adductor injuries occurred between the second measured session (week 9) and the third (week 15), where the addutor/abductor ratio was below 0.9. Further research is needed that compares ADD/ABD ratios above or below 0.90 and those that are stable vs. variable throughout the season.

Some previous studies indicated that decreased muscle strength in the hip adductors may be an important risk factor for groin injuries in football players [14,16,17]. In fact, our TEAM A showed a significantly decreased ADD power of 50% in the dominant leg between the third (week 15) and fourth (week 21) measured sessions, with subsequent injuries between the measured fourth (week 21) and fifth (week 27) sessions. The decrease in ADD power in the dominant leg for TEAM A could be explained by a reduction in the training volume for this leg after 2 weeks off in the Christmas period, while TEAM B only had 6 days off. The strength training applied to TEAM A was based on a low training volume for the strong leg. In this group, the dominant leg was always stronger than the non-dominant leg from the first assessment, being feasible to prescribe a lower training volume for the adductor muscles in this leg. However, in TEAM B the adductor injuries occurred between the measured second session (week 9) and the third (week 21), where the power of both ADDs did not change. To the best of our knowledge, there is no prior study analyzing the effect of supplementary ADD and ABD training on a professional football team during a full season. Jensen et al. [9] performed 8 weeks of strength training of the hip adductors using elastic bands as an external load, which increased the ADD strength by 14% and ABD by 17%, concluding that this could have implications as a promising approach in the prevention of groin injuries in football. However, our results showed a decrease in ADD power in the dominant and non-dominant legs was partially associated with a subsequent adductor injury only in the group of elite football players in TEAM A. Future studies could increase the number of measured sessions (e.g., to weekly) to avoid a possible loss of power and its potential influence as a risk factor for adductor injuries.

The present study had several limitations. We only obtained results for isometric hip adduction and hip abduction tests before the pre-season. Usually, coaches at the professional level are concerned about negative side effects from testing during the season [18,19]. This test was performed by medical staff, and they decided that they did not need more measures. Our functional measures were correlated with the isometric test, and thus the results could be estimated. There was no control group. This is usual during interventions with professional football players, where it is very difficult not to offer the same training and attention from staff to all players during an intervention [18].

As practical applications, present study showed that an individualized strength training based on the functional test proposed can help coaches through monitoring the adductor and abductor strength in professional football players in order to optimize performance and minimize the risk of adductor injury.

## 5. Conclusions

The strength training programs proposed can help to prevent adductor muscle injuries in top professional football players. Both adductor injury prevention programs resulted in a very low rate of adductor injury and only mild-to-moderate severity throughout the season.

## Figures and Tables

**Figure 1 ijerph-17-06408-f001:**
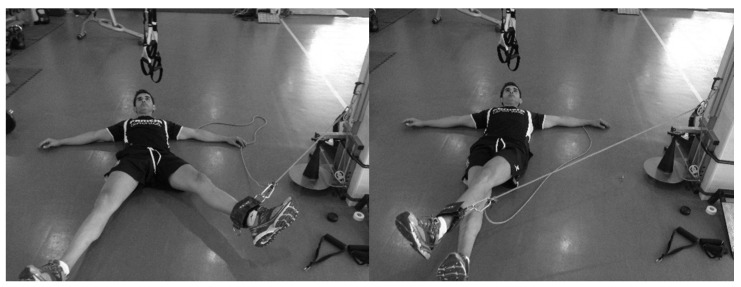
Rotary inertial device adductor exercise.

**Figure 2 ijerph-17-06408-f002:**
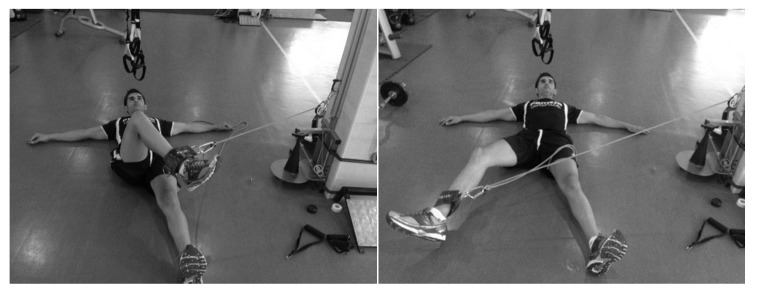
Rotary inertial device abductor exercise.

**Figure 3 ijerph-17-06408-f003:**
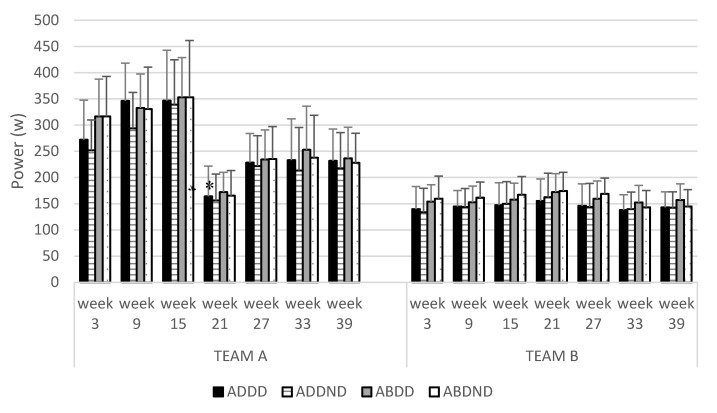
Mean power output of adductor and abductor exercises during the season. ADDD: Adductor dominant leg; ADDND: Adductor non-dominant leg; ABDD: Abductor dominant leg; ABDND: Abductor non-dominant leg (*) Significant differences with the last measure.

**Figure 4 ijerph-17-06408-f004:**
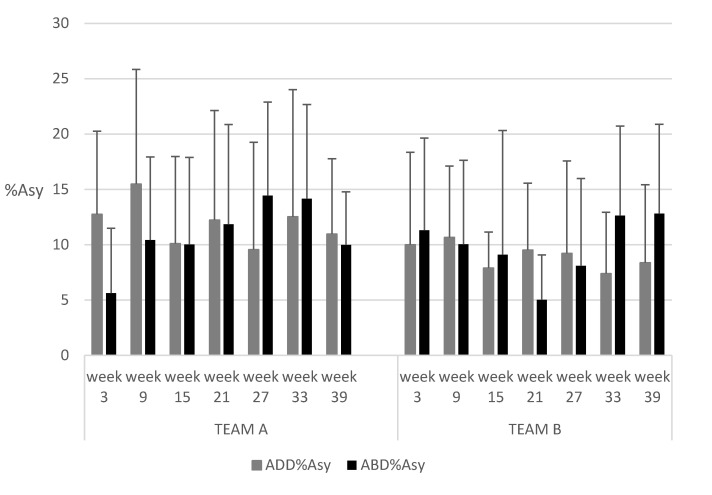
Evolution of ADD and ABD percentage of asymmetries between dominant and non-dominant legs (%Asy) during the season. ADD%Asy: ADD percentage of asymmetries between dominat and non-dominant legs; ABD%Asy: ABD percentage of asymmetries between dominant and non-dominant legs.

**Figure 5 ijerph-17-06408-f005:**
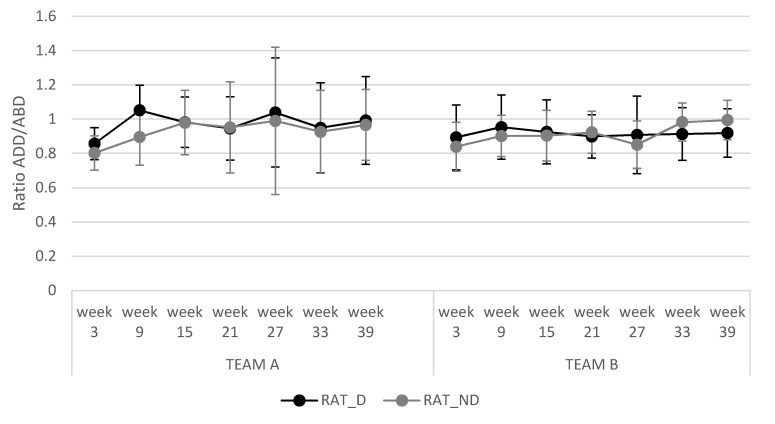
Evolution of adductor/abductor ratio during the season. RAT_D: adductor/abductor ratio dominant leg; RAT_ND: adductor/abductor ratio dominant leg.

**Table 1 ijerph-17-06408-t001:** The adductor injury prevention program volume (sets) based on unilateral deficit (%Asy) and adductor/abductor ratio (ADD/ABD ratio) output during the last measurement for TEAM A and TEAM B.

Conditioning Factors	TEAM A				TEAM B			
	Leg with Lower Power		Leg with Higher Power		Leg with Lower Power		Leg with Higher Power	
	ADD	ABD	ADD	ABD	ADD	ABD	ADD	ABD
%Asy < 10%	3	3	3	3	3	3	3	3
%Asy between 10.1 and 12.5%	3	3	2	2	4	4	3	3
%Asy between 12.6 and 15%	3	3	1	1	5	5	3	3
%Asy > 15.1%	3	3	0	0	6	6	3	3
	ADD/ABD ratio < 0.9							
%Asy < 10%	3	4	3	4	3	4	3	4
%Asy between 10.1 and 12.5%	3	4	2	3	4	5	3	4
%Asy between 12.6 and 15%	3	4	1	2	5	6	3	4
%Asy > 15.1%	3	4	0	1	6	7	3	4
	ADD/ABD ratio > 0.9							

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
