# Peer review of "Strength Conditioning Program to Prevent Adductor Muscle Strains in Football: Does it Really Help Professional Football Players?"

_ijerph, 2020, doi:10.3390/ijerph17176408_

Round 1

Reviewer 1 Report

GENERAL COMMENTS REGARDING PAPER

This study has a well done theoretical approach and it can be instructional for practitioners and researchers. However, there are some issues that have not been addressed in the manuscript. Despite such issues if these are addressed adequately it may be acceptable for publication following a second review.

MAJOR COMPULSORY REVISIONS

METHOD

MAJOR: pg 3, line 93: Please, report the accuracy of handheld dynamometer.

MAJOR: pg 4, line 143: How was the table 1 determined? Provide details to the reader.

MAJOR: pg 5, line 160: Based on the purpose of the present study, the standard error of measurement (SEM) should be reported for all measurements. Please, report them (absolute and relative and not use “ICC” to determine the “SEM”, because the ICC is already reported). It is so important for your experimental design. For detais, see Weir JP. Quantifying test-retest reliability using the intraclass correlation coefficient and the SEM. J Strength Cond Res. 2005 Feb;19(1):231-40.

MAJOR: pg 5, line 161: How was the CV determined? Provide details to the reader.

RESULTS

MAJOR: pg 5, line 168: Your findings look great, but the figures should be improved to be more interesting for the readers.

DISCUSSION

MAJOR: pg 8, lines 287-288: What are the practical applications?

Author Response

GENERAL COMMENTS REGARDING PAPER

This study has a well done theoretical approach and it can be instructional for practitioners and researchers. However, there are some issues that have not been addressed in the manuscript. Despite such issues if these are addressed adequately it may be acceptable for publication following a second review.

Answer: Authors would like to thank reviewer his/her labor on the manuscript at this first stage of the review process. We have substantially amended the paper to clearly present the methods to the reader. We hope we have clearly stated our  results, findings and conclusions.

MAJOR COMPULSORY REVISIONS

METHOD

MAJOR: pg 3, line 93: Please, report the accuracy of handheld dynamometer.

Answer: Thanks for your contribution. We re-phrase it, so that it now reads:”… using a portable handheld dynamometer (Lafayette Instrument Company, Indiana, USA; accuracy ± 0.88 N; reliability: ICC: 0.76 (0.24-0.94), CV: 8.7% and 0.89 (0.57-0.97), CV: 7.2% for hip-abduction and adduction respectively 21)  that was calibrated before testing.”

MAJOR: pg 4, line 143: How was the table 1 determined? Provide details to the reader.

Answer: Thank you for your consideration. We have generated Table 1 to provide the maximal details to the reader. As indicated in the text: "The training volume for ADD and ABD between monitoring sessions was determined by the last % Asy and ADD: ABD ratio obtained. The adductor injury prevention program for TEAM A was based on a reduced volume (i.e. number of sets ) for the leg with the higher power output, and in TEAM B was based on an increased volume for the leg with the lower power output ". Over a work of 3 sets of ADD and ABD with the dominant and non-dominant leg in the case you have a % Asy <10% and an ADD: ABD ratio <0.9, TEAM A decreases and TEAM B increases the number of sets for the leg with the higher and lower power respectively in the different possible scenarios (ie ADD: ABD ratio <0.9 or> 0.9 and % Asy between 10.1-12.5%, 12.6-15% and> 15.1%)

MAJOR: pg 5, line 160: Based on the purpose of the present study, the standard error of measurement (SEM) should be reported for all measurements. Please, report them (absolute and relative and not use “ICC” to determine the “SEM”, because the ICC is already reported). It is so important for your experimental design. For detais, see Weir JP. Quantifying test-retest reliability using the intraclass correlation coefficient and the SEM. J Strength Cond Res. 2005 Feb;19(1):231-40.

Answer: Thanks for your contribution. We do it, so that it now reads: in 2.7. Statistical Analysis ”Test-retest absolute reliability was assessed using the coefficient of variation (CV) and the standard error of measurement (SEM) (Hopkins, W. G. (2017). Spreadsheets for analysis of validity and reliability. Sportscience, 21.), whereas relative reliability was calculated using the intra-class correlation coefficient (ICC) with a 95% CI.” And in Results “Hip ADD/ABD exercises showed a good absolute and relative reliability for all measures: ICC: 0.92 (0.86-0.96); SEM: 0.30 watts; CV: 7.1%.” 

MAJOR: pg 5, line 161: How was the CV determined? Provide details to the reader.

Answer: Thanks for your contribution. We added the cite used to determine the CV, so that it now reads: ”Test-retest absolute reliability was assessed using the coefficient of variation (CV) and the standard error of measurement (SEM) (Hopkins, W. G. (2017). Spreadsheets for analysis of validity and reliability. Sportscience, 21.).

RESULTS

MAJOR: pg 5, line 168: Your findings look great, but the figures should be improved to be more interesting for the readers.

Answer: Thanks for your contribution. We have substantially amended the figures to clearly present the results to the reader

DISCUSSION

MAJOR: pg 8, lines 287-288: What are the practical applications?

Answer: Thanks for your contribution. We added a paragraph at the end of the discussion, so that it now reads: “As practical applications, present study showed that an individualized strength training based on the functional test proposed can help coaches by monitoring the adductor and abductor strength in professional soccer players in order to optimize performance and minimize the risk of adductor injury”.

Reviewer 2 Report

General remarks:
The paper concerns effectiveness of hip adductor injury prevention programs applied in football players. The topic is important and may attract readers. The article is generally clearly written, however there are some important issues which should be clarified. I had serious doubts when analyzing results presented in figure 3. I have never seen such a sudden decline in muscle power which dropped by half. Especially that it happened in one team. Maybe I am wrong but I suppose the authors made a mistake while measuring power: either equipment has been changed or measurement conditions. This needs to be explained. On the other hand authors declared they performed the mixed-design ANOVA. I have not found F and p values neither for GROUP factor nor repetition. The results presented suggest also a presence of a hudge interaction between those two factors what has not been reported. I would also expect partial eta squared as effect size for ANOVA to be presented.
Discussion is well written, however it mainly concerns injury rates. The second conclusion is not a consequence of the current results but rather tells about authors opinion supported by literature studies and experience.

Detailed remarks:
I suppose you did not read carefully [10, 12, 14]. Authors of those papers proposed values of variables which testified signs of recovery from injury - quite opposite to your statement about risk factors.
There are no descriptions of charts' axes as well as units in which results are expressed. There are no SDs in figures 3 and 5 it may be confusing.

I expect good explanation about results of teem A (fig.3)

Author Response

General remarks:
The paper concerns effectiveness of hip adductor injury prevention programs applied in football players. The topic is important and may attract readers. The article is generally clearly written, however there are some important issues which should be clarified. I had serious doubts when analyzing results presented in figure 3. I have never seen such a sudden decline in muscle power which dropped by half. Especially that it happened in one team. Maybe I am wrong but I suppose the authors made a mistake while measuring power: either equipment has been changed or measurement conditions. This needs to be explained. On the other hand authors declared they performed the mixed-design ANOVA. I have not found F and p values neither for GROUP factor nor repetition. The results presented suggest also a presence of a hudge interaction between those two factors what has not been reported. I would also expect partial eta squared as effect size for ANOVA to be presented.
Discussion is well written, however it mainly concerns injury rates. The second conclusion is not a consequence of the current results but rather tells about authors opinion supported by literature studies and experience.

Answer: Authors would like to thank reviewer his/her labor on the manuscript at this first stage of the review process. We have substantially amended the paper to clearly present the results to the reader.

Thank you for your consideration about the results showed in Figure 3. We explained this results in the discussion paragraph, so that it now reads”The decrease in ADD power in the dominant leg for TEAM A could be explained by a reduction in the training volume for this leg after two weeks off in the Christmas period, while TEAM B only had six days off. The strength training applied to TEAM A was based on a low training volume for the strong leg. In this group the dominant leg was always stronger than the non-dominant leg from the first assessment, being feasible to prescribe a lower training volume for the adductor muscles in this leg”.

Thank you for your consideration about the results of ANOVA. We are very sorry there has been an error in the description of the statistical method used. We have modified the statistical section, so that it now reads:” A repeated-measures design, using a one-way within-subjects ANOVA test, followed by Bonferroni post hoc tests, was used to investigate differences in variables measured within groups. A t-test was used to investigate differences in variables measured between groups.” Equally we modified the Results, so that it now reads:” Within-group analyses of consecutive monitoring training sessions showed a significant interaction for ADDD power, ADDND power, ABDD power, ABDND power ( F= from 8.647 to 11.428; p<0.001). Post-hoc analyses showed that TEAM A exhibited a significant reduction in ADDD power (ES= 3.43, p< 0.01), ADDND power (ES= 1.52, p< 0.01), ABDD power (ES= 3.65, p< 0.01) and ABDND power (ES= 2.20, p< 0.01) between the third (week 14) and fourth (week 20) monitored training sessions (See Figure 3)”.

Thank you for your consideration about the second conclusion. According to your proposal, we delete this conclusion.

Detailed remarks:
I suppose you did not read carefully [10, 12, 14]. Authors of those papers proposed values of variables which testified signs of recovery from injury - quite opposite to your statement about risk factors.

Answer: Thanks for your contribution. We re-phrase it, so that it now reads:” To date, adductor injury recovery factors have been suggested to include: a) an adductor/abductor ratio of more 90% 10 12 14; b) a lower extremity strength deficit of more than 10% on the injured compared with the uninjured leg 1 12; and c) a reduced hip adduction force with respect to the last measured force 10 14-17. Therefore, an injury prevention program should include exercises for hip adduction-abduction strengthening and testing in order to decrease the stress on the adductor muscle–tendon unit to prevent overuse injury 9.”

There are no descriptions of charts' axes as well as units in which results are expressed. There are no SDs in figures 3 and 5 it may be confusing.

Answer: Thanks for your contribution. We modified the Figures.

I expect good explanation about results of teem A (fig.3)

We answer before

Reviewer 3 Report

Interesting study presenting an injury prevention program for professional soccer players. Good design and presentation of both the Method and the Results. In the Discussion, the authors, in my opinion, rigorously address the most relevant aspects of the findings found in the context of the field of knowledge. Furthermore, some of the limitations of the study, such as the lack of a Control Group, have been indicated by the authors themselves, also pointing out the difficulty of having a control group in professional soccer.

In my opinion I have only seen one aspect that could improve the manuscript. The reviewed bibliography is not fully updated. In this sense, the authors only use 4 references from 2016 onwards, and it would be advisable to update the review; Very interesting manuscripts have been published (e.g. Clarsen group, among others) in recent years (2018, 2019 and 2020) that could enrich the Introduction, and especially the Discussion

Author Response

Interesting study presenting an injury prevention program for professional soccer players. Good design and presentation of both the Method and the Results. In the Discussion, the authors, in my opinion, rigorously address the most relevant aspects of the findings found in the context of the field of knowledge. Furthermore, some of the limitations of the study, such as the lack of a Control Group, have been indicated by the authors themselves, also pointing out the difficulty of having a control group in professional soccer.

Answer: Authors would like to thank reviewer his/her labor on the manuscript at this first stage of the review process.

In my opinion I have only seen one aspect that could improve the manuscript. The reviewed bibliography is not fully updated. In this sense, the authors only use 4 references from 2016 onwards, and it would be advisable to update the review; Very interesting manuscripts have been published (e.g. Clarsen group, among others) in recent years (2018, 2019 and 2020) that could enrich the Introduction, and especially the Discussion

Answer: Thanks for your appreciation. We have substantially amended the paper to actually the references.  We have searched for articles posted by Benjamin Clarsen and his group and they all address the topic of "Groin Injury prevention", but our goal was to analyse two adductor injury active prevention programs. They seem similar processes, but they are not, since one deals with a chronic injury (i.e. Groin Injury) and ours an acute injury (i.e. adductor injury), although in both the strengthening of the adductor muscles is a determining factor. If you consider that there is a more up-to-date article on this topic, we would be happy to introduce and discuss it in our study.

Harøy J, Clarsen B, Wiger EG, et al. The Adductor Strengthening Programme prevents groin problems among male football players: a cluster-randomised controlled trial. Br J Sports Med. 2019;53(3):150-157. doi:10.1136/bjsports-2017-098937

Harøy J, Pope D, Clarsen B, et al. Infographic. The Adductor Strengthening Programme prevents groin problems among male football players. Br J Sports Med. 2019;53(1):45-46. doi:10.1136/bjsports-2018-099993

Harøy J, Clarsen B, Thorborg K, Hölmich P, Bahr R, Andersen TE. Groin Problems in Male Soccer Players Are More Common Than Previously Reported. Am J Sports Med. 2017;45(6):1304-1308. doi:10.1177/0363546516687539

Round 2

Reviewer 1 Report

MAJOR: pg 5, line 160: Based on the purpose of the present study, the standard error of measurement (SEM) should be reported for all measurements. Please, report them (absolute and relative and not use “ICC” to determine the “SEM”, because the ICC is already reported). It is so important for your experimental design. For detais, see Weir JP. Quantifying test-retest reliability using the intraclass correlation coefficient and the SEM. J Strength Cond Res. 2005 Feb;19(1):231-40.

Answer: Thanks for your contribution. We do it, so that it now reads: in 2.7. Statistical Analysis ”Test-retest absolute reliability was assessed using the coefficient of variation (CV) and the standard error of measurement (SEM) (Hopkins, W. G. (2017). Spreadsheets for analysis of validity and reliability. Sportscience21.), whereas relative reliability was calculated using the intra-class correlation coefficient (ICC) with a 95% CI.” And in Results “Hip ADD/ABD exercises showed a good absolute and relative reliability for all measures: ICC: 0.92 (0.86-0.96); SEM: 0.30 watts; CV: 7.1%.” 

R2: It's not clear; SEM (absolute) = 0.30 watts and SEM (relative, i.e., SEM/Mean * 100) = 7.1%. If not, please provide the SEM relative and the last, what is the impact of this result on your prescription (Table 1.).

Author Response

R2: It's not clear; SEM (absolute) = 0.30 watts and SEM (relative, i.e., SEM/Mean * 100) = 7.1%. If not, please provide the SEM relative and the last, what is the impact of this result on your prescription (Table 1.).

15255

Answer: Thanks for your consideration. We provide the relative SEM as Typical error expressed by coefficient of variation (%CV) (Hopkins WG. Spreadsheets for analysis of validity and reliability. Sportscience. 2017;21). The impact of this result on our prescription is that it is considered that there is a modification of the %Asy from obtaining an approximate value of 1.5 times the% CV (% Asy = 10.1). As you can see, we offer a very progressive decrease / increase in the workload of the strong leg (TEAM A) and weak leg (TEAM B) respectively. The first interval of slight modification (+ 33% of the work volume) is established before a% Asy change between 1.4- 1.8 times the% CV (i.e. between 10.1-12.5 %Asy). The second interval of moderate modification (+ 66% volume) between 1.8-2.1 times the% CV (i.e. between 12.6-15 %Asy). And the third most severe modification interval (100% volume) above 2.1%CV (i.e above 15.1%Asy). We do not know if it is the best possible progression of work, but we do guarantee that to increase the volume there is a substantial change and that said change implies a modification greater than the minimum appreciable change for said valuation.

Reviewer 2 Report

Dear Authors,

Of course, I accept your explanation concerning surprising effect of the sudden drop of power. I am still wondered, but I should not oppose while I had not taken part in the measurement. Thank you for considering all my minor remarks. The paper looks better, now. Concerning statistics: I do not press, but that mixed design would be a better solution. Anyway, I accept the present approach. By the way, Cohen's d is used rather for dependent samples. The same indicator for between group comparisons is Hedges' g which has the same interpretation as Cohen's d. You can add it in the methods part.

Regards and good luck

Author Response

Of course, I accept your explanation concerning surprising effect of the sudden drop of power. I am still wondered, but I should not oppose while I had not taken part in the measurement. Thank you for considering all my minor remarks. The paper looks better, now. Concerning statistics: I do not press, but that mixed design would be a better solution. Anyway, I accept the present approach. By the way, Cohen's d is used rather for dependent samples. The same indicator for between group comparisons is Hedges' g which has the same interpretation as Cohen's d. You can add it in the methods part.

Answer: Thanks for your consideration. We modify the statistical method.
